# Evolution of Food and Nutrition Policy: A Tasmanian Case Study from 1994 to 2023

**DOI:** 10.3390/nu16070918

**Published:** 2024-03-22

**Authors:** Sandra Murray, Fred Gale, David Adams, Lisa Dalton

**Affiliations:** 1School of Health Science, University of Tasmania, Launceston, TAS 7250, Australia; 2School of Social Sciences, University of Tasmania, Launceston, TAS 7250, Australia; fred.gale@utas.edu.au; 3Tasmanian School of Business and Economics, University of Tasmania, Launceston, TAS 7250, Australia; david.adams@utas.edu.au; 4School of Health Science, University of Tasmania, New Norfolk, TAS 7140, Australia; lisa.dalton@utas.edu.au

**Keywords:** nutrition policy, food security, social justice, food systems, Tasmania, Australia, participation, community agency, sustainability, case study

## Abstract

Food security is a concept with evolving definitions and meanings, shaped by contested knowledge and changing contexts. The way in which food security is understood by governments impacts how it is addressed in public policy. This research investigates the evolution of discourses and practices in Tasmanian food and nutrition policies from 1994 to 2023. Four foundational documents were analysed using qualitative document analysis, revealing persistent food insecurity issues over three decades. The analysis identified a duality in addressing the persistent policy challenges of nutrition-related health issues and food insecurity: the balancing act between advancing public health improvements and safeguarding Tasmania’s economy. The research revealed that from 1994 to 2023, Tasmania’s food and nutrition policies and strategies have been characterised by various transitions and tensions. Traditional approaches, predominantly emphasising food availability and, to a limited extent, access, have persisted for over thirty years. The transition towards a more contemporary approach to food security, incorporating dimensions of utilisation, stability, sustainability, and agency, has been markedly slow, indicating systemic inertia. This points to an opportunity for future policy evolution, to move towards a dynamic and comprehensive approach. Such an approach would move beyond the narrow focus of food availability to address the complex multi-dimensional nature of food security.

## 1. Introduction

Food security is a concept with evolving definitions and meanings, and it is shaped by contested knowledge and changing contexts [1,2,3]. The way in which food security is understood by governments significantly influences how it is incorporated into public policy [4,5]. Despite efforts to ensure consistent access to food, traditional approaches focusing on stable access to affordable, nutritious, and culturally appropriate food have not succeeded in achieving food security for everyone. Global food security remains unachieved for all, as evidenced by the Food and Agricultural Organization (FAO) reporting that 30% of the global population were moderately or severely food-insecure in 2022 [6,7].

FAO defines food security as existing “when all people, at all times, have physical, social, and economic access to sufficient, safe, and nutritious food that meets their dietary needs and food preferences for an active and healthy life” [8], and this is acknowledged by the Organisation for Economic Cooperation and Development (OECD) [9]. They have persistently called for global action to address food security, as recognised in the UN Sustainable Development Goals (SDGs). SDG 2 sets targets to end hunger, achieve food security, improve nutrition, and promote sustainable agriculture by 2030 [10]. 

Food insecurity is recognised as a dangerous global problem that requires international cooperation and considered policy reform [11]. However, the development of comprehensive food and nutrition policy in a prosperous nation like Australia that operates within a liberal democratic federation and political system [12] has historically oscillated between national and state levels. Despite efforts, Australia has yet to achieve the goal of universal access to safe, nutritious, and adequate food for all of its citizens [13,14], making food security a pressing issue for individual states, like Tasmania, to address [15,16].

Australia’s first National Food and Nutrition policy, endorsed in 1992, emphasised the need for healthy diets and obesity and diet-related disease prevention and called attention to environmental sustainability and social justice considerations [17]. Such food and nutrition policies are nested within a much larger range of policies that all connect with the food system and often work at cross purposes. The focus of this research was to understand how the discourse of food and nutrition has evolved over the past 30 years within this policy sector, using one Australian state as a case study.

Tasmania is geographically separated from other Australian states. There is increasing concern among State health professionals as they grapple with the prominence of diet-related chronic diseases amid increased prosperity in globalisation and international trade with industrial agriculture and global productivity gains. This is accompanied by an increasing awareness of the social determinants of health, and a greater awareness of the intersection between food and the environment. Tasmania faces a strained health system and has some of the poorest health indicators in the country [18]. The reasons for these health indicators are multifaceted; however, some pertinent exacerbating factors include income disparities, limited educational attainment, and constrained geographic and affordable access to healthy diets [19,20,21]. Recent surveys indicate that between 22% and 51% of Tasmanians struggle to access healthy food [22,23]. Consequently, Tasmania continues to experience high and rising rates of chronic diseases linked to dietary patterns [24]. In response, Tasmanian governments have published food and nutrition policies and strategies iterations spanning from January 1994 to June 2023.

The evolution of food security discourse and policy frames in Tasmanian food and nutrition policy and strategies were investigated over this 30-year period. This paper describes the qualitative document analysis methodology, details the framing of food and nutrition problems and the evolution of food security policy and strategies, and then concludes with a discussion of future policy development opportunities to consider the social inclusion and empowerment of local communities.

## 2. Materials and Methods

This paper is part of a larger project exploring the practice of food justice in two local communities in an Australian state. This single-case-study methodology, focusing on Tasmania, involved five-step qualitative document analysis of five key Tasmanian food and nutrition policy documents. The steps included (i) document collection and extraction, (ii) establishing an analysis framework, (iii) document analysis, (iv) interpretation, and (v) reflexivity. 

### 2.1. Document Collection and Extraction

Food and nutrition policy documents published by the Tasmanian government between January 1994 and June 2023 were collected. Document inclusion and exclusion criteria (Table 1) were used to categorise the collected documents. 

A combination of systematic and snowball search strategies, involving a two-step process, was used to source relevant policy documents [26]. The first step involved searching the grey literature using Google Advanced and the key terms “food”, “nutrition”, “policy”, and “strategy”. Domain parameters were limited to websites of the Tasmanian Department of Health and Tasmanian Department of Premier and Cabinet, and internet archives. The second step involved the examination of reference lists of all retrieved documents for additional documents. 

Identified policy and strategy documents were systematically arranged in chronological order in terms of food and nutrition policy development in Tasmania. Key events associated with each policy release were included in the timeline and recorded at the state, national, and international levels. 

### 2.2. Establishing a Qualitative Analysis Framework

The interpretative policy analysis framework applied Braun and Clark’s [27,28] thematic analysis and Bacchi’s [29,30] “What’s the problem represented to be?” (WPR) approach. This combination enabled identification of latent themes and patterns in the selected documents, pertaining to food and nutrition frames, investigation of how the policies and strategies represent food and nutrition problems that they purport to address, and how governing takes place through these problematisations. Assumptions underlying the framing and representation of food security issues in Tasmanian over time, and whose voice was given prominence, or were absent or silenced in the policy process, were analysed using a combination of socio-ecological and empowerment theories which provided insights into the dynamics between personal, cultural, and structural influences on policy framing [31,32,33,34]. The integration of these theories into transition theory and the multi-level perspective (MLP) emphasises the importance of empowerment and participatory engagement in achieving systemic change [35,36,37]. This comprehensive analytical framework offers a nuanced understanding of the complex interplay between policy, practice and societal norms.

### 2.3. Document Analysis

All primary documents were read and re-read (SM) to acquire the data. The title, year, government in power, government department, nature of the document—i.e., policy or strategy—vision and goal, principles, evidence base, target audience and priority groups, action areas, partners, and actors were extracted (Appendix A). The documents were uploaded to NVIVO™ (version 12) and inductively coded using a line-by-line sequence.

Thirty-six codes and eleven categories were identified and then sorted into seven emergent themes, and patterns within the data were reviewed to ensure that they fit within the emergent themes, before four findings were named (Table 2). 

### 2.4. Interpretation

The corpus of documents were thematically analysed (SM, LD) to identify categories and patterns in meaning and to examine their relationships to each other, using Bacchi’s [29,30] six questions: identifying policy problems, presuppositions and assumptions, the effects of how the problems were represented and how problem representations were produced, disseminated, and defended. Socio-ecological theory [33,34] underpinned the findings discussion to consider the multi-level perspectives (MLPs) across personal, cultural, and structural domains [35,36].

### 2.5. Reflexivity

Rigor was assured in three ways. Team reflexivity and open discussions between the researchers in relation to their professional backgrounds in public health nutrition, public policy, government, and positionality ensured consideration of how these lenses may have influenced the analysis and research findings. For instance, two members of the team (DA, SM) possess both theoretical expertise and hands-on experience in the practical implementation of the policies being studied, lending significant authority to insights within this field. Coding discrepancies were resolved through full-team discussions, and all authors confirmed the final categories, themes, and interpretations in the findings (SM, FG, DA, LD). An audit trail documented all data collection and analysis decisions to enhance the study’s trustworthiness [38].

## 3. Results

Prior to detailing the two main findings, this section introduces a chronology of the Tasmanian Government’s food and nutrition policy agenda to show the complexity of the policy environment in which the evolution of discourse emerged within Tasmanian food and nutrition policy and strategy throughout the period studied. The first key finding highlights the emergence of two predominant food and nutrition problem frames. The second key finding details the transitions and tensions in food and nutrition policy and strategies from 1994 to 2023, revealing a duality in addressing the persistent policy challenge in nutrition-related health issues and food insecurity: the balancing act between advancing public health improvements and safeguarding Tasmania’s economy.

### 3.1. Chronology of the Tasmanian Government’s Food and Nutrition Policy Agenda

This research presents a detailed analysis of the Tasmanian government’s food and nutrition policy agenda across various administrations, including the historical context, problem frames, and policy shifts over the period from 1994 to 2023. Figure 1 provides a chronology of the Tasmanian Government’s food and nutrition policy and strategy iterations and their alignment to pertinent national and international agendas from 1992 to 2023, highlighting the complexity of the policy environment in which the evolution of discourse emerged within the food and nutrition sector.

The policy document search identified twenty documents, with no duplicates excluded and three deemed irrelevant. The remaining seventeen documents were subject to a full-text assessment for inclusion eligibility, resulting in thirteen determined as ineligible based on the criteria in Table 1. Four documents were ultimately included in the database for data extraction (Table 3). These comprised two ‘policies’ released in 1994 and 2004, both titled “Tasmanian Food and Nutrition policy” [39,40], that set guiding principles for understanding food and nutrition problems and solutions. Two more ‘strategy’ documents were also included: the 2012 “Food For All Food Security strategy” [41] and the 2021 “Food Relief to Resilience Food Security strategy” [42]. These strategies were intended to convey the Tasmanian government’s plan to ensure that households and communities had reliable access to an adequate and nutritious food supply.

### 3.2. Food and Nutrition Problem Frames in Tasmanian Policies and Strategies

The examination of food and nutrition problem frames identified two consistent themes including nutrition-related health issues and food insecurity. In 1994, nutrition-related health issues were presented as the primary policy problem, attributed to four main factors: “*diet-related illnesses, dietary intakes, alcohol and breastfeeding rates*” (DCHS (1994) [39], p. 22). The dietary habits were linked to seven diet-related illnesses, including “*coronary heart disease, cancer, cerebro-vascular disease and dental disease, eating disorders, and thyroid disease*”. (DCHS (1994) [39], p. 22)

This early policy problem frame also highlighted the significant influence of geographic and socioeconomic disparities, pointing to the crucial issues of equity and access to nutritious food as exacerbating factors in health inequalities, highlighting issues such as the following: 

“*geographical and social class gradients in diet-related conditions such as heart disease” were identified as “issues of equity and access to healthy food which is likely to contribute to worsening of these gradients*”. (DCHS (1994) [39], p. 63)

Nutrition challenges were again acknowledged in the 2004 policy, with added emphasis on poor nutrition causing ‘*significant individual and national cost*’ (DHHS (2004) [40], p. 9). This policy also identified that there was “*evidence of food insecurity among sections of the population*” (DHHS (2004) [40], p. 33) and defined food security as

“*the ability of individuals, households and communities to acquire food that is sufficient, reliable, nutritious, safe, acceptable and sustainable*”. (DHHS (2004) [40], p. 33)

In doing so, the 2004 food policy identified food insecurity as an area of concern warranting focused attention. By 2012, the food and nutrition strategy centralised the problem of food insecurity and continued to emphasise issues of social vulnerability in reporting that

“*as many as one in ten adults living in households with incomes in the bottom 20% of the total population experience food insecurity on a regular basis*”. (TFSC (2012) [41], Foreward)

The 2012 food policy was framed within a

“*socially inclusive policy perspective … [that] focuses on vulnerable people and places in addition to the more usual aspects of food security such as access to food, affordability, good nutrition, building resourcefulness and resilience in communities*”. (TFSC (2012) [41], p. 7)

The 2012 food policy “*focuses on vulnerable people*” (TFSC (2012) [41], p. 7) and offered a broader scope of food security than

“*the more usual aspects of food security such as access to food, affordability, and good nutrition, to include building resourcefulness and resilience in communities*”. (TFSC (2012) [41], p. 7)

In 2021, the food strategy centralised the vulnerability of marginalised groups and placed an emphasis on their inability to access and afford nutritious food and to make nutritionally conducive food choices, indicating that

“*food insecure households coped by eating less food and eating lower quality food, and only five per cent of food insecure respondents accessed emergency food relief distributors, indicating community food solutions to food insecurity remain a priority*”. (DOC (2021) [42], p. 6)

The strategy reported an acute increase in demand for food relief services and called for

“*pathways to increase community awareness and responsibility for food relief, including avenues for donation of in-kind support, such as backyard, community gardens or surplus produce*”. (DOC (2021) [42], p. 19)

### 3.3. Food and Nutrition Policy and Strategy Transitions and Tensions, 1994 to 2023

To address the two food and nutrition problem frames of nutrition-related health issues and food insecurity, the Tasmanian policies and strategies reflected various transitions and tensions. Two main policy agendas were framed, including public health and the state economy. These agendas competed in terms of whose interests were being served.

#### 3.3.1. Food Solutions for Public Health Improvements

Over the past 30 years, there has been an enduring commitment to social justice; however, the frames for developing solutions for achieving a more equitable food system have changed over time. With the shift from food and nutrition policies in 1994 and 2004 to strategies in 2012 and 2021, there was a concurrent shift from medical to social determinants of health in the agenda. The medical model was the dominant frame for early policy solutions, with subsequent strategies attempting to address the social determinants influencing food availability and access among vulnerable populations. 

##### Food Policy Origins: The Medical Agenda

In the 1994 and 2004 policies, food and nutrition were framed with a concern for public health vulnerability. Food solutions were entrenched in a medical model of health focused on physical and biological aspects of disease and illness, with an emphasis on population-based food and nutrition challenges: 

“*Obesity is a major public health concern as it contributes to ill health through a number of mechanisms. Government has a clear mandate for the promotion of public health and the achievement of social justice*”. (DCHS (1994) [39], p. 26)

The explicit medical agenda was to reduce the incidence of diet-related disease and multi-morbidities:

“*Tasmanians experience a significant burden of preventable diet-related chronic disease and food borne illnesses … (with) rates of heart disease, obesity, diabetes, hypertension, and some cancers as high as, and in some instances higher than other Australian States*”.(DHHS (2004) [40], p. 2)

Although couched in health language, the identified dietary problems followed a reductionist approach by addressing disease and illness through isolating and focusing on individual nutrients rather than considering the broader, holistic aspects of diet and health. Thus, the dietary problems were linked to

“*high sugar intake, high alcohol intake, high salt intake, high energy intake, high saturated fat intake, high fat intake, and low fibre intake*”.(DCHS (1994) [39], p. 22)

The policy corpus established a dual emphasis: firstly, on the need to influence the “*diet of the whole community*” (DHHS (2004) [40], p. 2), and, secondly, acknowledging the existence of differences and variability in the dietary composition among various Tasmanian population groups that might make them more nutritionally vulnerable than others.

##### Transition from Policies to Strategies: The Social Determinants in the Health Agenda

Within the 2012 and 2021 food security strategies, the link to diet-related chronic health problems was no longer highlighted and there was a transition towards a social health agenda. The new goal was to

“*achieve better food security outcomes for people and communities most at risk; embed responses to food security in policy and program development; build local food systems for community wellbeing and economic development*”. (TFSC (2012) [41], p. 12)

The resulting social health agenda provided an overarching framework for understanding the social determinants of health and food security to improve public health and reduce health disparities:

“*Through collaborative leadership and innovation we can all support Tasmanians in need to become more food secure, to improve the health and wellbeing of all Tasmanians*”. (DOC (2021) [42], pp. 12, 18)

##### Gesturing towards Social Justice

The 1994 and 2004 policies predominantly focused on food availability, while gesturing towards food access and social justice. The 1994 policy recognised “*access and social justice*” (DCHS (1994) [39], p. 66) as key issues that should not be concentrated “*solely on education but also address structural changes that will make healthy choices easier choices*” (DCHS (1994) [39], p. 11).

The two policies acknowledged the importance of all Tasmanians having access to food that is safe, nutritious, and affordable. However, there was a tension within the initial food policy document in which people were made responsible for their own health: “*all Tasmanians are in a position to choose a healthy diet*” (DCHS (1994) [39], p. 14).

By 2004, the framing of the food and nutrition policy had shifted to prioritise social justice within the context of food security, while simultaneously emphasising the importance of Tasmanian agricultural and environmental security. This dual focus was expressed in the policy vision:

“*Tasmania’s Vision for Food and Nutrition Tasmania: a State which produces quality, healthy, safe and affordable food, while sustaining the natural environment and strengthening the local economy; a community empowered to make food choices that enhance health and wellbeing*”.(DHHS (2004) [40], p. 3)

Whilst this dual vision worked to situate food and nutrition within a broad socio-ecological frame, there was a continued emphasis on the way in which some population groups were particularly vulnerable to food insecurity attributable to social inequality.

“*Most Tasmanians enjoy ready access to an ever-widening array of fresh foods, processed foods, ready-prepared foods and beverages, but a percentage of the population frequently worry about not having enough money to buy food for the household… Aside from financial barriers to accessing adequate food, some Tasmanians experience geographical, cultural and other social barriers*”. (DHHS (2004) [40], p. 2)

A new policy tension surfaced, in that whole communities were made responsible for making dietary improvements for the sake of their health. The 2004 policy called for 

“*a population approach that seeks to improve the diet of the whole community is likely to be more effective than working only with individuals who are seen to be at high risk of chronic preventable disease and food-borne illness*”.(DHHS (2004) [40], p. 2)

A subsequent shift occurred in the 2012 food security strategy away from a broader set of concerns expressed in the 1994 and 2004 policies to a more targeted plan to address food access issues. Here, language use changed from food insecurity, economic productivism, and personal responsibility to social inclusion, community food social enterprises, and rights. Note that access is emphasised, followed by availability, with the dominant production model prevalent and not challenged. Furthermore, the 2012 strategy clearly identified food access and supply as key determinants of food security. This access involved

“*the resources and capacity to acquire and use food such as transport to shops, financial resources, access to social eating environments, knowledge and skills about nutrition, and food choices*”. (TFSC (2012) [41], p. 6)

Supply involved “*production issues for growers, location of outlets, availability, price, quality, variety and promotion*”. (TFSC (2012) [41], p. 6)

With the issue of food security now more clearly defined and the determinants of food security succinctly identified, the focus of the 2012 Tasmania food security strategy was clearly articulated.

“*the focus of this strategy is on increasing access and supply of affordable and nutritious food and community driven approaches to food security for Tasmanians most at risk*”. (TFSC (2012) [41], p. 6)

Accordingly, the focus on a population-based approach to understanding food insecurity was refined with the use of specific social categories to identify the populations most at risk of being food-insecure.

“*people on low incomes, especially households dependent on government benefits and allowances; older people, especially those who are isolated or living alone; young people, especially children of single parent low income households; and isolated places, especially ‘food deserts’ where healthy food is difficult to get or absent*”. (TFSC (2012) [41], p. 6)

Four overarching priorities to address food insecurity at a local level were articulated:

“*increase food access and affordability; build community food solutions; regional development and support food social enterprises; and plan for local food systems*”. (TFSC (2012) [41], p. 6)

The 2021 food security strategy coincided with the COVID-19 pandemic as whole communities entered enforced lockdowns. Acknowledging that the lockdown period had perverse implications for people’s ability to access nutritious food, the Tasmanian government’s goal was to build resilience into the food relief system. This approach enhanced the role of Tasmania’s corporate productivist agrifood sector as it had become the source of the surplus that could be redistributed to food relief organisations to assist people “in need”. This was articulated as the need to

“*achieve an integrated food relief sector that supports Tasmanians in need to access sufficient, safe, nutritious, quality food, and access services that support long-term food resilience*”. (DOC (2021) [42], p. 16)

To achieve community resilience, the government identified the need to transition the food system from food relief to food resilience:

“*the strategy also shifted the broad language of ‘food system’ to adopt the more targeted language of the ‘food relief system’, comprising funding, food rescue, food relief and clients*”. (DOC (2021) [42], p. 10)

Communities, whether place or groups, were reaffirmed as being critical to achieving the new strategic goal with the implication that the ‘food relief community’ would be consulted or involved to ensure that food relief was distributed appropriately:

“*critical to this goal is the knowledge and expertise our communities hold about local need—and fostering the strengths and resources of state-wide and local organisations*”. (DOC (2021) [42], p. 3)

#### 3.3.2. Food Solutions for the State Economy

Despite acknowledging the importance of community engagement, the food policy consultation processes have not sufficiently included a diversity of voices, which means that food solutions have tended to prioritise the interests of food industry sectors. Alongside public health solutions, food solutions are framed within a corporatist business agenda, in which Tasmania is recognised as food export hub, considered critical to the state’s economy. 

##### Advancing Food Relief Solutions through Corporatist Business Partnerships

The 1994 and 2004 policy documents called for food insecurity to be addressed through sectoral and private industry partnerships. In this early policy era, the consultation process enabled people with a diverse range of interests in food and nutrition to attend key stakeholder forums:

“*the draft Policy was placed on a community consultation website along with a feedback guide. The consultation process was advertised in newspapers and a letter was sent to key stakeholders advising them of the consultation process and how they could become further involved*”. (DHHS (2004) [40], p. 8)

It is unclear how widely community members and consumers were engaged; however, consultation was reported to have occurred across a broad range of stakeholders and their representativeness.

“*There is a vast range of people with a specific interest in food and nutrition in the government, non-government, community and private sectors in Tasmania. These include farmers, food manufacturers, food retailers, food handlers, food transport workers, hospitality and catering workers, health professionals, teachers, regulatory bodies, peak industry organisations, government and non-government organisations and of course the consumers*”. (DHHS (2004) [40], p. 8)

While the policy formation process may have had an interest in engaging stakeholders on food policy issues, representation was dominated by agricultural and food industry business sectors, with less representation from consumer, health, and education stakeholders.

The 2012 and 2021 strategy documents attempted to be more socially inclusive than previous policy processes and include references to community participation and engagement to be facilitated through institutional representation. 

“*Political, industry and community leadership are also important, specifically leaders who can see the world through the lens of the person and family in the community, rather than through the lens of a programs or service*”. (TFSC (2012) [41], p. 22)

It was apparent that the Tasmanian Food Security Council (TFSC) in 2012 included institutional representation and did not include the voices of diverse community members within the Council membership. Lay community members, and, in particular, the voices of the vulnerable groups that the policy intended to target, were not included in this phase of food policy formation.

The 2021 strategy also used a consultation approach that omitted individuals with lived experience of food insecurity by instead including:

“*the food relief sector, including individual food relief providers and Tasmanian Government agencies with financial or operational investment in food production and distribution*”. (DOC (2021) [42], p. 12)

By engaging institutions and organisations to represent community and individual interests, the initiatives that were to follow the 2021 food strategy involved the Tasmanian Government instead working with the 

“*food relief sector to co-design an Action Plan, that will identify the activities we will undertake to support the food relief sector and their community partners to continue their critical work*”. (DOC (2021) [42]—Forward, p. 3)

The emphasis on the need to form sectoral and private industry partnerships to progress food relief solutions established a corporatist political system of interest representation and policymaking, whereby corporate groups came together and negotiated food relief initiatives on the basis of their common interests. The emergence of the corporatist business frame in food strategy and action has challenged policy aspirations for achieving a more socially just food system in Tasmania.

##### A More Just Whole-of-Food System versus Protecting Tasmania’s Economy

The 1994 and 2004 policies used a traditional, multisectoral, whole-of-food system approach, with strategies spanning primary production, consumption, environment, and the state’s economy while gesturing towards local food systems. On one hand, the policies acknowledged the need for a food system that made affordable and quality food available: 

“*steps may be needed to ensure that the nutritional advantages of our high quality and safe food are available and affordable to local consumers*”. (DHHS (2004) [40], p. 37)

On the other hand, the policies identify the food system as a major employer and contributor to the economy, with the 2004 policy reflecting the impact of tax reform on food pricing, gene technology, and organic farming. The government considered it important to ensure that

“*… Tasmania has a primary produce sector that is economically vibrant and produces safe and quality food*”. (DHHS (2004) [40], p. 36)

To achieve this goal, the government called for

“*an appropriate balance between maximising profits through export markets and ensuring availability and access to foods that are Tasmanian produced*”. (DHHS (2004) [40], p. 37)

To meet the need for affordable, quality food and create a food system that could contribute to a viable state economy, the early food and nutrition policies attempted to target the needs of multiple food systems stakeholders within an overarching economic frame:

“*food production and manufacture are vital to the Tasmanian economy, and the fine food niche marketing and clean green reputation the State has interstate and overseas is particularly important. Sustaining this reputation places extra responsibilities on the food industry to maintain high standards of food safety and quality control*”. (DHHS (2004) [40], p. 2)

The Tasmanian Government has a long history of being tasked with implementing structural and redistributive changes at a food system level to protect Tasmania’s economic security. The initial food policy pointed to the importance of Tasmania being a food export hub:

“*a limited local market and an extremely productive agricultural sector has moulded Tasmanian agriculture into a highly export oriented industry*”. (DCHS (1994) [39], p. 33)

The economic value of food and its importance to Tasmania’s economy has been explicit:

“*food makes a substantial contribution to the economy of Tasmania by providing employment opportunities and export revenue*”. (DHHS (2004) [40], p. 2)

To balance the competing economic agendas of addressing food security to improve public health and addressing it to protect Tasmania’s economic interests in food, another important strategy transition occurred.

##### Food System Solution Transitions: Responsibilising Communities to Achieve Food Security

By 2012, the food security strategy had shifted to planning for local food system transformation at the community level.

“*It focuses on those aspects of food security in community control such as capacity building and local food systems rather than agriculture and aquaculture industry development and protection, water and irrigation schemes, and global forces*”. (TFSC (2012) [41], Forward)

In 2012, there was a social justice dimension to policy; however, it was nested within a broader set of policies that limited its capacity to be achieved. The social justice endeavour was

“*creating resilient and sustainable communities, what drives this and how food systems can be part of the solution. Local food systems are important because they enable people to contributee to their own wellbeing through localised sustainable solutions grounded in local contexts*”. (TFSC (2012) [41], p. 7)

In 2021, the subsequent food security strategy narrowed its emphasis and used a food assistance-centred approach, focusing on strengthening the food relief system to address immediate food relief needs. The objective was to

“*identify pathways to increase community awareness and responsibility for food relief, including avenues for donation of in-kind support, such as backyard, community gardens or surplus produce*”. (DOC (2021) [42], p. 19)

Accordingly, the 2021 food strategy responsibilised communities for food insecurity and thereby potentially absolved the Tasmanian Government and the Tasmanian agri-food sector from any responsibility for achieving food security. Instead, the corporatist business agenda could be more easily realised by seeking ways to grow the export sector to achieve Tasmania’s Agri vision goal to grow the annual value of the State’s agriculture to USD 10 billion in exports by 2050.

## 4. Discussion

This research investigated the evolution of Tasmanian food and nutrition policy and strategy over the last 30 years, uncovering significant transitions and tensions. The findings identify two persistent policy frames: the complex interplay of nutrition-related health issues and the multifaceted nature of food insecurity, underscored by transitions and tensions between public health improvements and economic considerations. 

### 4.1. Nutrition-Related Health Issues and the Medical Model of Health

Initially, the policy focus was centred on addressing nutrition-related health issues, with a strong emphasis on the medical model of health. This approach, while evidence-based [43], often overlooked broader social determinants of health that underpin health disparities, such as geographic location and socioeconomic concerns, contributing to disparities in equitable access to nutritious food. Public health research has consistently identified food as a crucial determinant of health [44,45]. Consequently, food insecurity emerges as the second persistent policy frame within Tasmanian food and nutrition policy and strategy.

### 4.2. Evolving Conceptualisation of Food Security

The framing of food security within Tasmanian policies and strategies has evolved to incorporate the four dimensions globally recognised in food security discourse: availability, access, utilisation, and stability. These dimensions collectively encompass the consistent availability of sufficient food; the accessibility of food through both physical and economic means; the appropriate use of food through adequate dietary intake, nutrition knowledge, and food preparation; and the assurance of food stability over time to prevent disruptions due to sudden shocks or cyclical events [8,46]. Yet, despite the acknowledgement of these dimensions, policy implementation remains narrowly focused on food availability, and, to a lesser extent, on food access.

The introduction of “agency” and “sustainability” by the High-Level Panel of Experts (HLPE) on Food Security and Nutrition (FSN) in 2020 [2,3] marked a significant shift towards empowering individuals and communities, advocating for a more sustainable and participatory approach to food systems. This evolution underscores the importance of incorporating social justice principles into policy development, highlighting the need for policies that respond effectively to the voices and needs of all stakeholders. The variations in Tasmanian food security frames are important because they influence how successive governments responded to the two policy problems over time [47].

### 4.3. Government Response and Policy Shifts

There has been a consistent commitment from successive governments to respond to the complex issues of nutrition-related health issues and food insecurity. Over the past 30 years, Tasmanian food and nutrition policies and strategies have had an enduring focus on social justice; however, the frames for developing solutions to achieve a more equitable food system have changed over time. With the shift from food and nutrition policy to food security strategy, there was a concurrent shift from medical to social determinants in the health agenda. It is worth noting that Tasmania’s inaugural food and nutrition policy emerged in 1994 against an backdrop of international efforts to combat hunger and malnutrition, including World Food Summits and global declarations advocating for the eradication of these issues [48]. Additionally, public policy was in the early stages of emerging within a social-determinants-of-health framework [44]. The medical model was the dominant frame for early policy solutions, with the subsequent food security strategies attempting to address social determinants of health influencing food access, availability, utilisation, and stability in vulnerable populations.

### 4.4. Socio-Ecological Model and Public Health

The socio-ecological model [33,34] offers a lens through which to view the relationship between individual, cultural, and structural factors that are inherent in understanding food insecurity. Earlier policies, including the 2004 Food and Nutrition policy and the 2012 Food Security strategy, adopted the FAO’s 2006 definition of food security, focusing on individual, household, and community access to sufficient and nutritious food [8]. In contrast, the 2021 food security strategy adopted the FAO’s earlier 1996 definition, focusing on universal access to sufficient, safe, and nutritious food. This policy direction focused on the supply and availability of food and sought to address food security problems through food relief and assistance from surplus food that had been rescued or donated [49]. Accordingly, food became a resource for the charitable food sector and an efficient way of solving several problems simultaneously while keeping the corporate food system on track to grow [50].

### 4.5. Economic and Health Dualities

Alongside concerns for improving public health, Tasmanian food and nutrition policy and strategies have grappled with supporting the state’s economy. Despite intentions to address food insecurity comprehensively, policies and strategies have often leaned towards economic resilience, at the expense of broader social and health outcomes. This focus is partly due to consultative processes that insufficiently represent the voices of people with lived experience of food insecurity, or who have professional expertise, leading to strategies that prioritise economic solutions, such as food relief through corporatist business partnerships. It is an approach that is sometimes criticised for framing food security primarily as an economic issue [14]; the 2021 strategy, intensified by COVID-19 challenges, shifted towards a systems-driven approach, aiming to build resilience in the food relief system with significant reliance on the corporate agrifood sector [49]. This evolution underscores a “regime lock-in”, where economic and industry priorities are so entrenched that they resist change and dominate the discourse by hindering environmental and social considerations, despite a changing global context, societal needs, and environmental challenges [35,36].

### 4.6. Local Systems and Global Contexts

Transforming Tasmania’s food system to create a more socially just and food-secure state has proven challenging. Recent strategies have recognised food as an important economic asset, positioning Tasmania as an export hub to protect the state’s economy. Transitioning food strategy to focus on improving local rather than whole-food systems can be considered as a niche radical innovation [35,36]. The most recent food strategy (2021), particularly during the COVID-19 pandemic, hinted at reducing reliance on global food system dependency, the benefits of which may promote equity, community engagement, cultural relevance, and economic and social justice [51,52]. Despite the intentions to address social and ecological challenges through localism, issues, such as stakeholder complexity, resource demands, the extensive coordination of policies and governance across multiple sectors, and equity concerns that may inadvertently disadvantage vulnerable communities as the food system undergoes changes, underline some of the systemic resistance to change within the dominant food security ‘regime’ [36,52,53]. 

### 4.7. Towards Agency and Participation

The persistent commitment to improve food security in Tasmania has consistently focused on improving both food availability and access. While earlier policies emphasised food availability, subsequent strategies have attempted to broaden their scope to prioritising the food security dimensions of access and introducing opportunities to build niche community-based solutions. This indicates a notable shift towards incorporating agency, with communities actively engaged in making decisions and influencing change. Conversely, the 2021 strategy’s emphasis on food relief highlights the persistent challenges in addressing structural factors impacting food security [54]. The evolution of policy from public–private partnerships to genuine community participation has lagged, reflecting the influence of the socio-technical landscape. The prevailing voice in earlier policies has been dominated by the agriculture sector, where economic priorities and institutional voices have historically overshadowed grassroots efforts. Despite attempts to engage communities in recent strategies, the consultation process has fallen short of representing mainstream community development principles for engaging individuals with lived experience. Community voice, knowledge, and wisdom have not been clearly apparent in the policy development process. Instead, the policy dialogue has tended to be shaped more by institutional interests [55,56]. 

### 4.8. Empowerment and Community Engagement

Contemporary approaches to food security recognise the fundamental role of agency for all individuals, including the voices of the most marginalised and those of limited financial means, underscoring the importance of adopting participatory and people-centred approaches in policy development [55,56]. This focus on enhancing individuals’ capacity to make choices and have a say in shaping the local food system, reinforced by the recent addition of agency as a dimension of food security [2,3], is pivotal to empowering communities to strengthen their own resilience [57,58]. By empowering individuals, households, communities, or identity-based groups, there is a transformative shift towards active participation and decision making in the food system rather than as passive recipients of government policies or donor assistance. 

In conclusion, this research suggests the next evolution of food and nutrition policy development towards adopting participatory and people-centred approaches, with a focus on empowerment and community engagement. Navigating the socio-technical landscape to rebalance power dynamics within food systems and enable communities to actively participate, while acknowledging a diversity of voices, emerges as a priority when developing future policies. These policies must adeptly address the intricate challenges of food insecurity [59,60], ensuring that all community members, especially those historically marginalised, have a meaningful role in shaping policies [61,62,63,64]. 

### 4.9. Limitations

The research highlights the limitations inherent to discourse analysis for capturing the practical impacts of these policies and strategies, underscoring the complexity of translating policy language into actionable outcomes. The presence of multiple, competing agendas within this contested domain further amplifies the challenge of advancing food and nutrition policy. For example, the socio-technical landscape that encompasses global environmental changes, economic shifts, and evolving social norms further complicates the advancement of food and nutrition policy. This broader landscape sets the context in which these policies operate, exerting additional pressure to the existing “regime lock-in” and demanding a more adaptive and responsive policy framework. Despite nominal shifts towards wider considerations since 1994, the absence of substantial policy transformation signals the need for a paradigm shift. This shift necessitates enhancing agency and empowering all stakeholders within the food system, particularly marginalised communities who are often resource-constrained.

## 5. Conclusions

Tasmania stands at a pivotal moment in the evolution of its food and nutrition landscape. This research reveals a persistent adherence to traditional approaches to food security, primarily focused on increasing food availability and, to a limited extent, access. This narrow perspective signifies a “regime lock-in”, where established policies and practices have resisted meaningful evolution, despite a growing recognition of the need for a broader, more nuanced approach to food security. Such an approach would encompass additional dimensions of food security—utilisation, stability, sustainability, and agency—moving towards a more integrated, contemporary model. 

Crucially, the incorporation of social justice principles, which are intrinsically linked to agency, into Tasmanian food and nutrition policies and strategies remains underdeveloped. This highlights an opportunity for marginalised community groups to actively participate in the policymaking process. A shift towards more inclusive policy development can facilitate the integration of community-led solutions, democratising food systems and challenging the prevailing “regime lock-in”.

Despite the introduction of broader food security dimensions, the dominant discourse has continued to prioritise food availability, signalling a persistent resistance to change within the policy regime. As the challenges faced by marginalised groups in Tasmania intensifies, adopting a transformative approach grounded in food justice becomes increasingly critical. The future direction of food security policy in Tasmania stands at a crossroads, pointing to an opportunity for future policy development to transition from its current static approach to a more dynamic and comprehensive approach, moving beyond the narrow focus of food availability to address the complex multi-dimensional nature of food security.

To transcend the current “regime lock-in” and initiate a more desirable new stage of evolution, a collective effort from all stakeholders is essential. This will entail dismantling the barriers to change and embracing a collaborative approach that spans personal, cultural, and structural levels. By navigating the socio-technical landscape with a focus on empowerment and participatory engagement, Tasmania can pave the way for a future where food security policies are responsive to both the nuanced needs of its communities and the broader environmental and social challenges. The path forward lies in cultivating an inclusive environment where food security encompasses the full spectrum of its multi-dimensional nature, responsive to both internal policy dynamics and broader socio-technical contexts.

## Figures and Tables

**Figure 1 nutrients-16-00918-f001:**
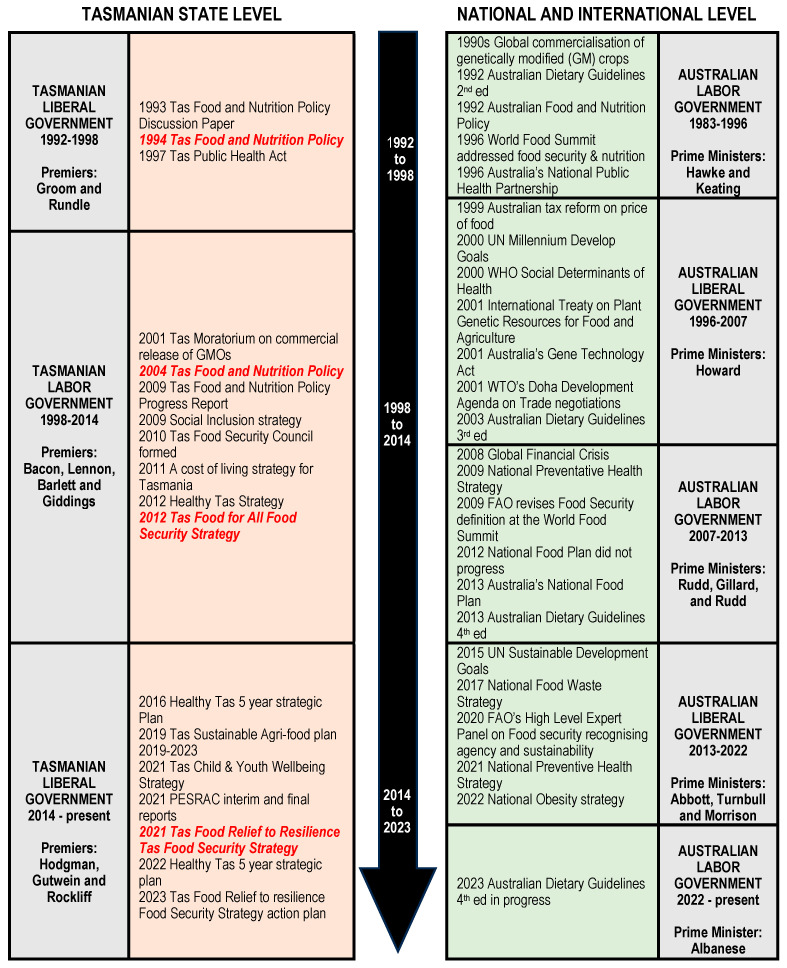
Chronology of the Tasmanian Government’s food and nutrition policy agenda and alignment with the national and international agenda, 1992 to 2023. (Abbreviations—Aust, Australia; PESRAC, Premier’s Economic and Social Recovery Advisory Committee; Tas, Tasmania; WHO, World Health Organisation; WTO, World Trade Organisation).

**Table 1 nutrients-16-00918-t001:** Inclusions and exclusion criteria for policy document analysis *.

Characteristics	Included Documents	Excluded Documents
Publication characteristics	English January 1994–June 2023 Full text available	Non-English Outside of time parameters Full text inaccessible
Publisher	Tasmanian Government	Food and beverage companies and industry groups Non-government organisations
Document type	Policies Strategies Action plans	Media releases, communiques, declarations, resolutions, speeches, submissions, briefs, Hansard, acts, reports, action plans, frameworks, discussion papers, white paper
Document content	Whole-population outcomes Addresses obesity, non-communicable diseases (NCDs), nutrition, or food Proposes/identifies specific policy actions or interventions	Outcomes are population-specific (e.g., children, maternal, population affected by disease) Addresses infectious diseases or other communicable diseases Documents fail to specify any policy actions or interventions

* Whilst Tasmanian legislation was initially considered for inclusion, this did not occur because there was no singular legislative instrument governing food security. Instead, food, nutrition, and food security are spread across several government portfolios, including Health Regulation, Health and Wellbeing, Planning, Economics, Tourism, and Sustainability [25].

**Table 2 nutrients-16-00918-t002:** Thematic analysis codes, categories, and emergent themes.

Codes	Categories	Emergent Themes
Nutrition Health and wellbeing Food and nutrition system	Diet-related disease. Disease prevention	Nutrition-related health issues combining a medical health agenda and social health agenda
Vulnerable populations Make healthy choices easier Promotion of healthy eating Nutrition education	Behaviour changes and personal responsibility	
Food production, supply, and availability Food technology Food industry and product manufacturing	Industrial agriculture	Whole food system combining industrial agriculture and economic growth
Food distribution retail wholesale Workforce development Economy Food marketing advertising Institution food service catering	Food system approach	
Mapping surveillance monitoring and measuring Governance Government leadership and decision making Food regulations	Procedural governance	
Environmental regulations	Environment	
Local food solutions Community place-based approach Regional development social enterprise	Local food system solutions	Local food system
Cost of living Social inclusion Community development Right to food and social justice	Social justice	Food security: availability, access, use, and stability
Food relief and distribution Agriculture Food sector Community position Food insecurity Food access affordability Food use Food stability Planning and procurement	Food availability, access, use, stability	Food relief Food business and corporatisation
Multi sectorial approach Partnership and collaboration with key stakeholders	Partnership and collaboration	Private and public sector partnerships
Community participation and codesign Food resilience Community leadership	Community participation	Food resilience

**Table 3 nutrients-16-00918-t003:** Policy documents included in the document analysis.

Year	Document Title	Responsible
1994	Tasmanian Food and Nutrition Policy	Tasmanian Department of Community and Health Services (DCHS)
2004	Tasmanian Food and Nutrition Policy	Tasmanian Depart of Health and Human Services (DHHS)
2012	Food for All Tasmanians—A Food Security Strategy	Tasmanian Food Security Council (TFSC)
2021	Food Relief to Food Resilience—Tasmanian Food Security Strategy 2021–2024	Tas Department of Premier and Cabinet (DPAC)

## Data Availability

The original contributions presented in the study are included in the Appendix A, further inquiries can be directed to the corresponding author.

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
