# Peer review of "Evolution of Food and Nutrition Policy: A Tasmanian Case Study from 1994 to 2023"

_nutrients, 2024, doi:10.3390/nu16070918_

Round 1

Reviewer 1 Report

Comments and Suggestions for Authors

General comments

It is an interesting paper, focusing on a very actual topic, such as food security.  Overall, it is quite a good and informative paper but a little difficult to follow (too many quotes-sections, 3.2.2.., 3.23 ) 

Minor comments

 The methodology section- is a little bit confusing to me the description of the whole process of searching and selecting documents with Google, with lots of details regarding mostly the excluded documents (documents mentioned in Table 1), while the criteria for inclusion seem to rely on document type (Reports Action plans Frameworks Discussion papers White paper) and the analysis is finally based on 20/17/ 4?? Policy documents published on the Tasmanian government website.  Also, Figure 1 is not very relevant as information is already presented in the text. More details related to the coding process, frequencies, the program used, emergent themes, and study limitations would be useful.

Figure 2 is very interesting, complex/detailed but difficult to follow and interpret by the wider reader / a simplified one, including only relevant documents/acts, will probably be helpful.

Section 3.1-easier to follow if you arrange in a table/frame different approaches/discourses

Author Response

Dear Reviewer 1,

Thankyou very much for your constructive feedback. I have attached a PDF document which explains how we have addressed your feedback.  It has taken us a little longer than anticipated to make the required changes and to ensure that feedback from each reviewer was included.

Kind regards

Reviewer 2 Report

Comments and Suggestions for Authors

This article analyses the succession of regulations that govern food security in Tasmania from 1994 to the present day. It is clear, well-organized and written. The corpus compiled and the method adopted are satisfactorily described. The results are essentially based on an analysis of each of the 4 founding texts and an identification of the evolution of the approach by comparing these texts with each other. The use of bibliographic references is relevant and quite comprehensive.

My assessment is favourable because this article brings new elements and helps to identify major challenges for food security. It highlights the increasing complexity of the ways in which food security framing and representations are enriched and transformed. The progressiveness of evolutions is a fairly frequent object of analysis in papers dealing with transitions. The main result is that these developments are disappointing and that the necessary transformations are far too slow. I felt that this article should be accepted with “minor modifications” which I detail below around 3 main suggestions to the authors.

1 - No reference is made to the Transition Theories, even though that is what it is all about. For structuring the discussion, the authors would have had tools for understanding and interpreting the developments analysed. In particular, the notion of “lock-in” would be useful in order to show how successive texts are indicative of the forms of resistance of the dominant “regime”. The island's productivism and export vocation are firmly established and dominate debates on food security. Any change that aims to introduce environmental and social issues is experienced as a tension between the status quo and the consideration of extra-economic criteria. The 4 titles within the 3.2 are highly significant of the tensions the authors identify clearly, with expression such as “with fact of” (3.1), the use of “while gesturing” (3.2, 3.3) and “with some experimentation” (3.4). These tensions and their consequences (in the texts) remain highly dependent on the political balance of power. Moreover, the analysis concerns only the discourse and not the practical impacts of the texts, and it could be relevant to draw the perspective of new studies including the “on-the-ground” effects.

The socio-technical "landscape" (an important element of the Transition theory) would have been useful to question: social movements and their ability to take up “hot societal issues” such as the greening of production techniques and support for organic farming, the ban of GMOs, the political dimensions of food relief, the ability allowed to the marginalized populations to have a voice. After reading the article, there is still a feeling of absence from public opinion (an island is a “micro-society” where people know well each other) and tensions of protest emanating from the population. And it remains difficult to understand the forces at work for the changes of the 4 texts: how and for what reasons the identified shifts are occurring?

2 – It seems to me that one element of the analysis is missing, that of putting developments at federal level into perspective. While Figure 2 compares the two situations of the State of Tasmania and the Federation of Australia, the authors no longer make use of this enrichment of the analysis. It would have been interesting to know what was happening at the federal level while at the state level new texts were being prepared and promulgated.

In particular, the various turning points identified and the corresponding room for manoeuvre in Tasmania would benefit from being compared with the movements made (or not) at the federal level. The discussion deserves to include an analysis of the respective “periodizations” of the State of Tasmania vs. the Federation of Australia, especially since the political balances and the political colorations of both the executives are often out of sync. Are there any advances or delays in the evolution between the two situations? Does one influence the other?

3 – The whole end of the discussion and even more so the conclusion are won by a “normative” and prescriptive posture. The evolutions initially suggested then become necessities (formulated as such) and the authors present them as having to be imposed on the policy-makers in order to be in line with history. It seems to me that this change of posture is problematic in a scientific article because it is largely based on the authors' convictions and not on the knowledge produced.

While generally agreeing with what is presented as absolute need, would it not be better for the authors to present it as the “desirable” new stage of evolution? And then ask themselves what conditions are required for it to happen. This would be much more useful than adopting the posture of a "lesson giver" and a prescriber of public policy, even a militant. Among these conditions required, it is clear that the various forces involved defend partially divergent interests and that it is becoming important to build more ambitious consensuses, promoting larger and faster evolutions. This is not decreed on the basis of rational arguments, but by acting on the situation so that these ideas eventually prevail. The changes expected in this final part are therefore rather rewording and rephrasing in order to better highlight how new regulatory texts could go further in the direction desired by the authors and how to overcome the obstacles to their design.

A few typos to be corrected:

L433 : “Its” to be replaced by “It is”.

L482 : “adopted a” is duplicated. One has to be cancelled.

L506 : Before “While” there is a “_” to be cancelled.

L510 : There is a “and” before “form” to be cancelled.

L515 : Replace “intensions” by “intentions”.

Author Response

Dear Reviewer 2,

Thankyou very much for your constructive feedback. I have attached a PDF document which explains how we have addressed your feedback.  It has taken us a little longer than anticipated to make the required changes and to ensure that feedback from each reviewer was included.

Kind regards
